# Novel Amperometric Biosensor Based on Tyrosinase/Chitosan Nanoparticles for Sensitive and Interference-Free Detection of Total Catecholamine

**DOI:** 10.3390/bios12070519

**Published:** 2022-07-12

**Authors:** Valeria Gigli, Cristina Tortolini, Eliana Capecchi, Antonio Angeloni, Andrea Lenzi, Riccarda Antiochia

**Affiliations:** 1Department of Experimental Medicine, Sapienza University of Rome, Viale Regina Elena 324, 00166 Rome, Italy; valeria.gigli@uniroma1.it (V.G.); cristina.tortolini@uniroma1.it (C.T.); antonio.angeloni@uniroma1.it (A.A.); andrea.lenzi@uniroma1.it (A.L.); 2Department of Biological and Ecological Sciences, University of Tuscia, 01100 Viterbo, Italy; e.capecchi@unitus.it; 3Department of Chemistry and Drug Technologies, Sapienza University of Rome, Piazzale Aldo Moro 5, 00185 Rome, Italy

**Keywords:** total catecholamine, tyrosinase, chitosan nanoparticles, amperometric biosensor, graphene screen-printed electrode

## Abstract

The regulation of nervous and cardiovascular systems and some brain-related behaviors, such as stress, panic, anxiety, and depression, are strictly dependent on the levels of the main catecholamines of clinical interest, dopamine (DA), epinephrine (EP), and norepinephrine (NEP). Therefore, there is an urgent need for a reliable sensing device able to accurately monitor them in biological fluids for early diagnosis of the diseases related to their abnormal levels. In this paper, we present the first tyrosinase (Tyr)-based biosensor based on chitosan nanoparticles (ChitNPs) for total catecholamine (CA) detection in human urine samples. ChitNPs were synthetized according to an ionic gelation process and successively characterized by SEM and EDX techniques. The screen-printed graphene electrode was prepared by a two-step drop-casting method of: (i) ChitNPS; and (ii) Tyr enzyme. Optimization of the electrochemical platform was performed in terms of the loading method of Tyr on ChitNPs (nanoprecipitation and layer-by-layer), enzyme concentration, and enzyme immobilization with and without 1-ethyl-3-(3-dimethylaminopropyl)-carbodiimide (EDC) and N-hydroxysuccinimide (NHS) as cross-linking agents. The Tyr/EDC-NHS/ChitNPs nanocomposite showed good conductivity and biocompatibility with Tyr enzyme, as evidenced by its high biocatalytic activity toward the oxidation of DA, EP, and NEP to the relative o-quinone derivatives electrochemically reduced at the modified electrode. The resulting Tyr/EDC-NHS/ChitNPs-based biosensor performs interference-free total catecholamine detection, expressed as a DA concentration, with a very low LOD of 0.17 μM, an excellent sensitivity of 0.583 μA μM^−1^ cm^−2^, good stability, and a fast response time (3 s). The performance of the biosensor was successively assessed in human urine samples, showing satisfactory results and, thus, demonstrating the feasibility of the proposed biosensor for analyzing total CA in physiological samples.

## 1. Introduction

Catecholamines (Cas) are both neurotransmitters and hormones that play fundamental roles in central nervous and cardiovascular systems [1,2,3]. High CA levels, in particular dopamine (DA), epinephrine (EP), and norepinephrine (NEP), indicate cardiotoxicity leading to tachycardia and heart failure [4]. In contrast, low levels of CA are responsible for several neurological diseases, such as Parkinson’s disease, Alzheimer’s disease, epilepsy, stress, and depression [5,6]. Therefore, accurate monitoring of these important biomarkers in serum, blood, and urine is very important in early diagnostics [7].

To address this issue, several analytical methods have been developed for in vitro CA monitoring, such as high-performance liquid chromatography (HPLC) [8], HPLC with tandem mass spectrometric detection [9,10,11], fluorescence [12], and field-effect transistor [13] methods. These methods show high sensitivity and selectivity but require sophisticated and expensive equipment, time-consuming procedures, and skilled operators. Electrochemical methods represent an interesting alternative thanks to their unique advantages, such as their low cost, rapid response, simple operation, and capability for in situ detection [14,15,16,17].

Unlike glucose, CAs are redox-active compounds that are easily oxidized in a potential window between 0 and 0.4 V vs. Ag/AgCl [18], depending on the electrode modification. Therefore, total CA concentration can effectively be measured without the use of redox mediators and/or enzymes. The direct oxidation of DA, EP, and NEP on different electrode platforms allowed for the development of several sensors for CA detection [19,20,21,22,23,24,25]. However, these sensors suffer from relatively high oxidation potential and electrode surface passivation values due to the electrogenerated phenoxy radicals [26,27]. To this end, attempts at the bio-catalytic oxidation of CA, in particular DA and NEP, have been realized by integrating the Tyr enzyme with conductive nanomaterials and/or polymers on the electrode surface [28,29], leading to significant increases in DA and NEP oxidation peaks with concomitant cathodic shifts in their peak potential. Nevertheless, signal interference from other biological molecules, such as ascorbic acid (AA) and uric acid (UA), which show similar oxidation potentials with overlapping voltametric signals, represents another critical issue for CA detection based on oxidation signals that may limit its applicability. To address this issue, Tyr-based biosensors for CA detection are mainly used by monitoring the reduction signal of the bio-catalytically produced quinone compounds at a relatively moderate potential (about 0.1 V vs. Ag/AgCl on a carbon electrode) [30,31,32], where no electrochemical interference signal is present. Nanoparticles and/or nanostructures have been integrated into different sensing platforms to enhance both biosensor sensitivity and biosensor selectivity toward CA detection [32,33,34,35,36,37,38,39,40].

Tyrosinase (Tyr, polyphenol oxidase, EC 1.14.18.1) is a binuclear blue copper protein that acts as a polyphenol oxidase [41]. It catalyzes two consecutive oxidation reactions in presence of molecular oxygen, showing both monophenolase and diphenolase activity: (i) the o-hydroxylation of phenols to guaiacol; and (ii) the oxidation of guaiacol to o-quinones. The biosensor principle is the measurement of the increase in the cathodic signal due to the electrochemical reduction of the o-catecholamine quinone involving two electrons and two protons (Reaction 2), enzymatically generated on the electrode surface by Tyr (Reaction 1), according to the following reactions:Tyr catecholamine + ½ O_2_ → o-catecholamine-quinone + H_2_O (1)
o-catecholamine-quinone + 2H^+^ + 2e^−^ → catecholamine + H_2_O(2)

The signal amplification is assured by the recycling of the o-catecholamine quinone by Tyr enzyme, leading to an enhancement of the selectivity and sensitivity of the biosensor. Therefore, the key factor in biosensor construction is adequate and effective Tyr immobilization in order to enhance the catalytic activity. Common approaches utilized for Tyr immobilization include carbon paste immobilization [42], sol–gel immobilization [43], entrapment within electropolymerized conducting polymers [37], and physical adsorption [44]. A considerable amount of attention has been paid to the use of chitosan (Chit), a polysaccharidic biopolymer composed of glucosamine with proper surface functional groups for biological/chemical binding and/or rapid adsorption [45,46]. It has found a great deal of applicability in sensing applications because of its non-toxicity, biocompatibility, biodegradability, and low cost [45]. Thanks to its polycationic character, chitosan can lock negatively charged proteins by charge attractions and hydrogen-bond interactions. Moreover, the presence of reactive amino and hydroxyl groups in its linear structure allows for covalent protein immobilization through the coupling chemistry via glutaraldehyde or EDC/NHS cross-linking agents [47,48]. However, its poor electrical conductivity may limit its applicability. To overcome this issue, chitosan has generally been combined with nanomaterials, such as graphene [49], multi-walled carbon nanotubes [50], and metallic nanoparticles [20,34,36], as well as conducting polymers, such as polyaniline and polypyrrole [51,52]. Chitosan nanoparticles (ChitNPs) with a size smaller than 100 nm showed superior chemical and biological performances than pristine chitosan, making them an excellent source for biomedical and biotechnological applications [53,54]. Thanks to their high surface–volume ratio, they can be utilized in biosensing applications as enzyme immobilization supports to achieve ultrasensitive detection of biomolecules. To date, only a few reports have been published with this aim. Anusha et al. [55] immobilized glucose oxidase (GOx) over ChitNPs on a gold electrode and, subsequently, Kim et al. [56] developed a platform based on a poly-γ-glutamic acid/ChitNPs hydrogel, which embedded GOx and magnetic nanoparticles.

In this paper, we report a new biosensing platform for the effective detection of total catecholamine based on the immobilization of Tyr in ChitNPs by using the zero-length EDC-NHS cross-linker on a graphene screen-printed electrode. To the best of our knowledge, no reports are present in the literature on electrode platforms based on a ChitNPs/EDC-NHS/Tyr nanocomposite for CA detection.

## 2. Experimental

### 2.1. Chemicals and Reagents

Tyrosinase (EC 1.14.18.1, 8503 U mg^−1^ from mushroom, Tyr), dopamine hydrochloride (DA), epinephrine (EP), norepinephrine (NEP), chitosan (low molecular weight) (Chit), sodium tripolyphosphate (TPP), 1-ethyl-3-(3-dimethylaminopropyl) carbodiimide (EDC), N-hydroxysuccinimide (NHS), ascorbic acid (AA), and uric acid (UA) were purchased from Sigma-Aldrich (St. Louis, MO, USA). Phosphate buffer was prepared with Na_2_HPO_4_ and NaH_2_PO_4_. NaOH and HCl were used to adjust pH values. Graphene screen-printed carbon electrodes (GPH/SPEs), used for biosensor construction, were purchased from Dropsens (Metrohm Dropsense, Asturias, Spain, model DRP-110GPH, screen-printed carbon electrodes modified with graphene).

### 2.2. Apparatus

Electrochemical measurements were performed in a 10 mL conventional three-electrode thermostated glass cell (model 6.1415.150, Metrohm, Herisau, Switzerland) using a graphene screen-printed carbon electrode (GPH/SPE) as a working electrode, an external Ag/AgCl/KCl_sat_ electrode (198 mV vs. NHE) as a reference electrode (cat. 6.0726.100, Metrohm, Herisau, Switzerland), and a glassy carbon rod as a counter electrode (cat. 6.1248.040, Metrohm, Herisau, Switzerland). An Autolab Potentiostat/Galvanostat (Eco Chemie, The Netherlands) was utilized in the electrochemical measurements. Scanning electron microscopy (SEM) and energy-dispersive X-ray spectroscopy (EDX) measurements were performed with a High-Resolution Field Emission Scanning Electron Microscope (HR FESEM, Zeiss Auriga Microscopy, Jena, Germany). The chronoamperometry experiments were carried out using a Sensit/SMART portable potentiostat (PalmSens, Houten, The Netherlands), a smartphone-based sensing device directly connected to a smartphone for POC signal reading.

### 2.3. Synthesis of Chitosan Nanoparticles

Chitosan nanoparticles (ChitNPs) were synthesized according to an ionic gelation process by adding 9 mL of chitosan dispersion (1 mg mL^−1^) to 3 mL of TPP solution (1 mg mL^−1^) and stirring the solution for 2 h at 25 °C with gentle magnetic stirring until an opalescent solution formed. The solution was successively centrifuged for 15 min at 6000 rpm, and the precipitated pellets were rinsed with DI water, dried, and used for further characterization [57].

### 2.4. Fabrication of TYR/ChitNPs-Modified Electrodes

Two different approaches were employed to immobilize Tyr and ChitNPs on a graphene SPE (the layer-by-layer (LbL) method and the nanoprecipitation (Np) method).

#### 2.4.1. Layer-by-Layer Method

The layer-by-layer method (LbL) consists of the direct deposition “layer-by-layer” of each component of the electrochemical platform (ChitNPs and Tyr) onto the electrode. LbL-Tyr/ChitNPs/GPH/SPE was prepared as follows:(1)A total of 6 μL of ChitNPs solution (1 mg mL^−1^ in a water solution) was drop-cast onto GPH/SPE and left to dry at room temperature in air for approximately 1 h;(2)A total of 6 μL of Tyr solution (0.5 mg mL^−1^ in PBS 0.1 M, pH 7.2, KCl 0.1 M) was drop-cast onto the GPH/SPE previously modified with ChitNPs and left to dry at room temperature in air for approximately 1 h.

In the case of LbL-Tyr/EDC-NHS/ChitNPs/GPH/SPE, an additional electrode modification step was introduced between Step 1 and Step 2 by drop-casting onto the electrode surface a mixture of 3 μL of EDC solution (0.5 mM in PBS 0.1 M, pH 7.2, KCl 0.1 M) and 3 μL of NHS solution (0.1 mM in PBS 0.1 M, pH 7.2, KCl 0.1 M).

Three electrodes of each type were prepared in order to evaluate the reproducibility of the biosensor.

#### 2.4.2. Nanoprecipitation Method

In the nanoprecipitation method (Np), Tyr and ChitNPs are co-precipitated in the same solution. Briefly, a solution of ChitNPs (1 mg mL^−1^) was added to a solution of Tyr (0.5 mg mL^−1^) under gentle magnetic stirring for 2 h at room temperature and subsequently deposited onto the electrode surface. The Np-Tyr/ChitNPs/GPH/SPE was prepared by drop-casting 12 μL of the Tyr–ChitNPs solution onto the GPH/SPE, which was then dried at room temperature in air for approximately 1 h.

Three Np-Tyr/ChitNPs/GPH/SPEs were prepared in order to evaluate the reproducibility of the biosensor.

### 2.5. Electrochemical Measurements

Cyclic voltammetry (CV) experiments were performed in PBS at a scan rate of 10 mV s^−1^. Magnetically stirred PBS (0.1 M, pH 7.2, KCl 0.1 M) was used to perform amperometric measurements. Electrochemical impedance spectroscopy (EIS) was performed at the open circuit potential (OCP) without bias voltage in the 0.1–10^4^ Hz frequency range using an ac signal with an amplitude of 10 mV. Differential pulse voltammetry (DPV) experiments were recorded from −0.6 to 0.6 V with an amplitude 20 mV and a step potential of −5 mV. Baseline corrections were done for all DPV data using the NOVA software.

### 2.6. Preparation of Human Urine Samples

Human urine samples were diluted 100-fold with PBS (0.1 M, pH 7.2; KCl 0.1 M) before all measurements.

## 3. Results and Discussion

### 3.1. Characterization of ChitNPs

The morphological analysis of ChitNPs was realized by field emission scanning electron microscopy (FE-SEM) and energy-dispersive X-ray spectroscopy (EDX). Figure 1 shows the FE-SEM images of a bare GPH/SPE before (Panel A) and after (Panel C) the modification with ChitNPs. The bare GPH/SPE showed a densely packed rough surface, consistent with previous literature [58]. The EDX spectrum confirmed the presence of only carbon and oxygen, with an atomic percentage, calculated from the quantification of the peaks, of about 98% and 2%, respectively (Figure 1, Panel B). The FE-SEM image of ChitNPs/GPH/SPE shows aggregates of spherical nanoparticles with different diameters ranging from 70 to 300 nm (Figure 1, Panel C). The presence of the ChitNP aggregates may be ascribed to the inter- and intra-molecular cross-linkages mediated by the interaction of the positively charged amino groups of Chit and the negatively charged TPP ions [59]. The EDX spectrum of ChitNPs/GPH/SPE (Figure 1, Panel D) confirmed the presence of C (96%), O (3%), and a lower amount of Na (0.5%), probably due to the use of the TPP cross-linker in the procedure for the synthesis of ChitNPs.

All GPH/SPEs and ChitNPs/GPH/SPEs were subsequently characterized by cyclic voltammetry (CV) experiments in a solution of Fe(CN)_6_^4−^ (data not shown) in order to calculate the electroactive area (A_EA_), the heterogeneous electron transfer rate constant (K_0_, cm s^−1^), and the roughness factor (electroactive–geometrical area ratio, *ρ*), which are reported in Appendix A. The A_EA_ was evaluated using the Randles–Sevick equation by the slope of the peak current vs. the square root of the scan rate (v^1/2^) [60]. The K_0_ was calculated using the extended method, which merges the Klingler–Kochi and Nicholson–Shain methods for totally irreversible and reversible systems, respectively [61,62].

### 3.2. Electrochemical Characterization of the Tyr/ChitNPs Biosensor Platform

The electrochemical characterization of the Tyr/ChitNPs platform was performed using cyclic voltammetry (CV) and electrochemical impedance spectroscopy (EIS).

#### 3.2.1. Optimization of Methods for Loading Tyr on ChitNPs

Two different methods for loading Tyr on the ChitNPs/GHP/SPE platform were investigated by CV experiments: (i) the nanoprecipitation method (Np); and (ii) the layer-by-layer (LbL) method. Figure 2 shows the cyclic voltammograms of DA in the absence (ChitNPs/GPH/SPE, black line) and the presence of Tyr enzyme immobilized by the Np method (Np-Tyr/ChitNPs/GPH/SPE, red line), and the LbL method (LbL-Tyr/ChitNPs/GPH/SPE, blue line) in comparison with Tyr enzyme immobilized on a simple layer of unstructured chitosan (Tyr/Chit/GPH/SPE, green line) deposited by a drop-casting method. Two well-defined redox peaks, due to the oxidation/reduction of DA, are clearly visible on the bare electrode (black curve) without Tyr, while no redox peaks are present with the Tyr/Chit/GPH/SPE electrode (green curve), attesting to the lower conductivity of the unstructured chitosan layer [45] compared with nanostructured chitosan. A similar result was obtained by recording the CVs of DA on a chitosan-modified electrode in the absence of Tyr enzyme (Chit/GPH/SPE). The cyclic voltammogram of DA on ChitNPs/GPH/SPE shows two clear redox peaks (Appendix A, black curve), while smaller oxidation and reduction peaks in the reverse scan can be observed with unstructured chitosan (Appendix A, red curve), thus confirming that the nanostructuration strongly improves the chitosan’s conductivity. Similar behavior was described by Marroquin et al. [63], who reported that chitosan nanocomposite films realized with Fe_3_O_4_ and MWCNTs could simultaneously have enhanced the mechanical properties, thermal stability, and electrical conductivity.

An increase in the reduction peaks was observed for both Np-Tyr/ChitNPs/GPH/SPE (Figure 2, red curve) and LbL-Tyr/ChitNPs/GPH/SPE (Figure 2, blue curve) due to the reduction in the dopamine-o-quinone produced by the enzymatic reaction on the electrode surface to DA [39], demonstrating that the immobilization of Tyr within the nano-biopolymer greatly enhanced the catalytic activity of the enzyme. The reduction peak current of DA on LbL-Tyr/ChitNPs/GPH/SPE was about three times higher than that on Np-Tyr/ChitNPs/GPH/SPE, indicating the superior electrochemical performances of the LbL loading method in terms of the better electron transfer between the dopamine-o-quinone and the electrode surface [64]. The results reveal that the LbL immobilization method promotes the Tyr’s catalytic property better than the Np method and probably plays a significant role in: (i) the improvement of enzyme immobilization by favoring physical adsorption and the formation of electrostatic interactions between positively charged ChitNPs [57] and negatively charged Tyr enzyme [41]; and (ii) facilitating the direct electron transfer between the substrate molecules and the modified electrode surface.

#### 3.2.2. Optimization of Tyr Concentration

The amount of immobilized enzyme is one of the most relevant parameters in electrode modification. Figure 3 shows the effect of the concentration of Tyr, immobilized by the LbL method on the Tyr/ChitNPs/GPH/SPE, on the cathodic peak current registered in the presence of 50 μM DA. The intensity current differences (∆I), calculated as the current differences obtained in the presence and the absence of Tyr, increase with decreasing concentrations of Tyr immobilized on the electrode surface. The optimal Tyr concentration was found to be 0.5 mg mL^−1^, attesting to the fact that electron transfer might be hindered at higher enzyme concentrations. Lower concentrations were also tested, showing no significant increase in the ∆I values.

#### 3.2.3. Tyrosinase Covalent Functionalization via EDC-NHS Coupling Chemistry

In order to enhance the efficacy of the Tyr-based biosensor, the Tyr covalent immobilization strategy using EDC-NHS as a cross-linking agent was investigated. The EDC-NHS coupling chemistry takes advantage of the properties of the EDC-NHS cross-linker, which is able to form amide bonds between carboxyl and amine groups, as shown in Figure 1 [48]. In particular, the carboxyl groups of ChitNPs immobilized on the electrode surface are activated by the EDC/NHS, forming a typical NHS-ester, and, subsequently, the amino group from the Tyr enzyme acts on this NHS-ester, forming a very strong covalent amidic linkage.

Figure 4 shows the cyclic voltammograms of DA in pH 7.2 PBS on ChitNPs/GPH/SPE (black line) and LbL-Tyr/ChitNPs/GPH/SPE without (blue line) and with (red line) the EDC-NHS cross-linking agent. The reduction peak current of DA observed on LbL-Tyr/EDC-NHS/ChitNPs/GPH/SPE was 1.5 times higher than that on LbL-Tyr/ChitNPs/GPH/SPE, attesting to the enhanced electrochemical performances of the method based on covalent functionalization compared with the LbL method utilizing simple electrostatic interactions, probably thanks to the higher stability of the covalently immobilized Tyr for enhanced biocatalysis.

#### 3.2.4. Electrochemical Impedance Spectroscopy Behavior of the Modified Electrodes

The EIS technique was used to investigate the electrochemical properties of the modified electrodes after each surface modification step. Figure 5 shows the Nyquist plots of the ChitNPs/GPH/SPE (black), the Tyr/ChitNPs/GPH/SPE (blue), and the Tyr/EDC-NHS/ChitNPs/GPH/SPE (red) in 5 mM [Fe(CN)_6_]^3−/4−^ solution. The semicircle diameter of the Nyquist plot represents the charge transfer resistance (Rct), which is related to small changes at the electrode–electrolyte interface in the solution. The immobilization of a protein biomolecule over the electrode surface causes an increase in the Rct value that is related to the hindering of the electron transfer between the redox probe and the electrode surface. The impedance spectra were fitted by using the simple Randles circuit (R(Q(RW))) reported in Figure 5. Experimental R_ct_ values are in the following order: ChitNPs/GPH/SPE (357 Ω) < Tyr/ChitNPs/GPH/SPE (403 Ω) < Tyr/EDC-NHS/ChitNPs/GPH/SPE (436 Ω) due to the hampering effect of Tyr on the charge transfer. The Rct values confirm the successful immobilization of Tyr enzyme on the electrode surface, resulting in strong optimization with the use of the EDC-NHS cross-linker (red curve).

### 3.3. Amperometric Response of Dopamine

Figure 6 (Panel A and B) illustrates a typical steady-state current–time response upon successive additions of DA on Tyr/ChitNPs/GPH/SPE (Panel A) and Tyr/EDC-NHS/ChitNPs/GPH/SPE (Panel B), showing a fast response time (3 s). The measurements were carried out by immersing the modified electrode in blank PBS (0.1 M). Once a steady baseline was achieved, DA was added to the solution, causing an increase in the cathodic current due to the reduction in o-quinone generated by the enzymatic oxidation of DA [64]. Thus, the target DA molecule was regenerated during the reduction reaction at the electrode, allowing for the amplification of the electrochemical signal, which leads to enhanced sensitivity [65]. Figure 6 (Panel C) exhibits the typical calibration curves of the DA biosensors corresponding to the electrode platforms with (black curve) and without (red curve) EDC-NHS, both following Michelis–Menten kinetics. As can be seen in the inset of Panel C, the linear range spanned from 4 μM to 12 μM with a regression equation of y = 0.03325x + 0.0404 (R^2^ = 0.994) and from 0.5 μM to 5 μM with a regression equation of y = 0.80066x + 0.0039121 (R^2^ = 0.997) in the case of the Tyr/ChitNPs/GPH/SPE biosensor and the Tyr/EDC-NHS/ChitNPs/GPH/SPE biosensor, respectively. The biosensor based on the EDC-NHS cross-linker showed a lower LOD of 0.17 μM and a two-fold higher sensitivity of 0.583 μA mM^−1^ cm^−2^ compared with the corresponding biosensor without EDC-NHS. According to the Lineweaver–Burk equation [66], the apparent Michaelis–Menten constants ($K_{M}^{\mathrm{app}}$) were 25.59 and 9.51 μM for the Tyr/ChitNPs/GPH/SPE biosensor and the Tyr/EDC-NHS/ChitNPs/GPH/SPE biosensor, respectively. It is interesting to note that the experimental $K_{M}^{\mathrm{app}}$ value is lower than the K_M_ value of the free enzyme towards dopamine (K_M_ = 2.2 mM) [67], attesting to the optimal accessibility of the target molecules to the active sites of the enzyme. The electrochemical characteristics of the two biosensors in terms of the $K_{M}^{\mathrm{app}}$, maximum current density, linearity range, LOD, and sensitivity are summarized in Table 1.

### 3.4. Amperometric Response of Epinephrine and Norepinephrine

The Tyr/EDC-NHS/ChitNPs/GPH/SPE-based biosensor, which showed the best performance, was utilized for EP and NEP detection. Chronoamperometric experiments were performed at an applied potential of 0 V vs. Ag/AgCl as EP and NEP gave cathodic peaks at the same redox potential of DA (curves not shown). The corresponding calibration plots are reported in Figure 7, together with the calibration plot of DA (black curve) as a comparison. Table 2 summarizes the analytical and kinetic performances of the proposed biosensor for the three catecholamines studied, including the linear range, LOD, sensitivity, maximum current density, and $K_{M}^{\mathrm{app}}.\;$The sensitivity of the DA biosensor was 5 times higher than that of EP and NEP. The $K_{M}^{\mathrm{app}}$ values were in the following order: DA < NEP < EP, attesting to the stronger affinity and binding of the Tyr enzyme to DA, leading to a higher catalytic activity [27].

The total catecholamine concentration is therefore expressed as a DA concentration, thanks to the best analytical and kinetic performance of this biosensor.

Table 3 shows a comparison between the results obtained with our biosensor and other Tyr-based biosensors for catechol and catecholamine recently reported in the literature. It is possible to observe that the proposed ChitNPs/graphene platform shows superior analytical performances compared with the other platforms for CA detection, whereas inferior performances can be observed when compared with Tyr-based catechol biosensors. These results can be ascribed to the stronger affinity of the Tyr enzyme to catechol.

### 3.5. Selectivity

The selectivity of the Tyr/EDC-NHS/ChitNPs/GPH/SPE-based biosensor for DA was investigated by recording the current response in the presence of potential interferents, which can coexist in biological matrices. In particular, differential pulse voltammograms were recorded in a solution containing a 100-fold concentration of ascorbic acid (AA) and uric acid (UA) in the presence of DA. AA and UA usually show overlapping voltametric curves on both unmodified and chemically modified electrodes [23]. As shown in Figure 8, three well-resolved cathodic peaks were recorded at about −450, +100, and +400 mV for AA, DA, and UA, respectively. The results clearly demonstrate that AA and UA did not significantly interfere with DA detection, showing the high selectivity of the proposed biosensor.

### 3.6. Stability

The stability and lifetime of the biosensor for DA, EP, and NEP detection were investigated by measuring the DPV responses of the biosensor every 5 days (10 measurements) over a period of 30 days using a 10 µM solution of each CA (Appendix A). The biosensor was stored in a dry refrigerator at 4 °C in between the measurements. The stability was good as the biosensor retained about 90% of its initial current response after 30 days. These results may be ascribed to the synergistic effect of the bio-nanocomposite including ChitNPs and graphene, which reduces the enzyme denaturation and the subsequent loss of catalytic properties.

### 3.7. Real Sample Detection

Finally, the Tyr/EDC-NHS/ChitNPs/GPH/SPE-based biosensor was tested in human urine samples spiked with known concentrations of DA in order to evaluate the effectiveness of the biosensor in physiological samples. A fixed amount of DA (5 μM) was added to three urine samples without any sample pretreatment. All measurements were done in triplicate using chronoamperometry and the results are reported in Table 4. The recovery rates ranged from 94% to 98% and the relative standard deviation (RSD) values did not exceed 6.3%, demonstrating the efficacy of the proposed biosensor in real samples.

The results were subsequently compared to those obtained with a standard spectrophotometric method used as reference (Table 3). An unpaired *t*-test was performed, and the results show that there was no statistically significant difference in the DA quantification between the proposed biosensor and the standard spectrophotometric method (t = −2429; degrees of freedom = 4; *p* = 0.072). This result indicates the biosensor’s excellent capacity to provide mean values close to those obtained when the same samples are measured by using the standard spectrophotometric method.

## 4. Conclusions

In this work, we demonstrated the feasibility of developing an amperometric biosensor for total catecholamine detection by using a novel electrochemical nano-platform based on the immobilization of Tyr/ChitNPs on a graphene SPE. The ChitNPs were synthesized by the ionic gelation process and subsequently immobilized with Tyr enzyme on the graphene electrode through a layer-by-layer approach with and without EDC-NHS as a cross-linking agent. The ChitNPs provided a friendly environment for Tyr immobilization, thus enhancing the catalytic activity of the enzyme and, at the same time, showing an improved platform conduction pathway thanks to the higher conductivity compared with pristine chitosan. The ChitNPs/EDC-NHS/graphene nanocomposite matrix showed superior electrochemical performances, which can be attributed to the synergistic effect of graphene and ChitNPs. The excellent conductivity and large surface area of the graphene electrode in combination with the properties of the ChitNPs produced the good analytical performances of the developed catecholamine biosensor in terms of a short response time (3 s), high sensitivity, a low detection limit, good stability, and satisfactory recovery when tested in spiked human urine samples. At the current state of research, the proposed ChitNPs/graphene platform does not provide superior performance compared with other nanostructured platforms for CA detection reported in the literature, but the ease of preparation and low cost of the proposed ChitNPs make them a promising component and/or additive for electrochemical biosensor platforms thanks to the described beneficial properties.

## Data Availability

Not applicable.

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
