# Peer review of "Novel Amperometric Biosensor Based on Tyrosinase/Chitosan Nanoparticles for Sensitive and Interference-Free Detection of Total Catecholamine"

_biosensors, 2022, doi:10.3390/bios12070519_

Round 1

Reviewer 1 Report

The authors presented an electrochemical enzyme-based biosensor using graphene, chitosan nanoparticles, and tyrosinase to detect total catecholamines. The paper can be published after applying the comments below. 

1. Several electrochemical parameters related to modified electrodes should have been reported and compared with a bare electrode such as The electroactive surface area (Aeas) and of the roughness factor (RF), well-known kinetic parameter (ψ), heterogeneous electron transfer constant (k0) of electrodes in a solution containing Fe(CN6)4-/3-.

2. The surface coverage of the immobilized enzyme (Гc) on the surface of the electrode, charge-transfer coefficient (α) and heterogeneous transfer rate constant (ks) should have been reported (10.1016/j.bioelechem.2011.04.004).

3. The effect of the Chit nanoparticles on the response of the biosensor should have been reported.

4. The t-test value and p-value of the response of the proposed biosensor should have been reported by taking into account of the response of the standard method to the same concentration.  

5. The applied potential was 0 V vs. Ag/AgCl but they did not show that the applied potential was optimized. 

Author Response

The answers to reviewer 1 are reported in the attached file

Reviewer 2 Report

The authors presented an interesting study on the development and characterization of a novel amperometric biosensor based on tyrosinase immobilized using chitosan nanoparticles for the detection of dopamine and other cathecolamines.

The manuscript presents good English level, but there some small mistakes and some unclear phrases in the text (see attached document). Please correct them.

The study was properly organized and described, with some possible improvements. In the attached document you have my remarks and suggestions for improving the manuscript. For these reasons, I recommend the publication of this manuscript only after minor revision.

Author Response

We made all the corrections suggested by Referee 2 in the attached document.

Round 2

Reviewer 1 Report

The authors replied to my comments well and I was satisfied with the answers. congratulation